

# Pandemic-driven changes in the nearshore non-commercial fishery in Hawai'i: catch photos posted to social media capture changes in fisher behavior

Timothy Grabowski[1], Michelle E. Benedum[2], Andrew Curley[3],
Cole Dill-De Sa[4] and Michelle Shuey[5]

[1] U.S. Geological Survey, Hawai'i Cooperative Fishery Research Unit, University of Hawaii at Hilo, Hilo, Hawai'i, United States
[2] Political Science, University of Colorado at Boulder, Boulder, Colorado, United States
[3] Anthropology Department, University of Hawaii at Hilo, Hilo, Hawai'i, United States
[4] Earth Systems Program, Stanford University, Stanford, California, United States
[5] Department of Geography and Environmental Sciences, University of Hawaii at Hilo, Hilo, Hawai'i, United States

Corresponding author
Michelle E. Benedum,
michelle.benedum@colorado.edu

## ABSTRACT

Using social media, we collect evidence for how nearshore fisheries are impacted by the global COVID-19 pandemic in Hawai'i. We later confirm our social media findings and obtain a more complete understanding of the changes in nearshore non-commercial fisheries in Hawai'i through a more conventional approach—speaking directly with fishers. Resource users posted photographs to social media nearly three times as often during the pandemic with nearly double the number of fishes pictured per post. Individuals who fished for subsistence were more likely to increase the amount of time spent fishing and relied more on their catch for food security. Furthermore, individuals fishing exclusively for subsistence were more likely to fish for different species during the pandemic than individuals fishing recreationally. Traditional data collection methods are resource-intensive and this study shows that during times of rapid changes, be it ecological or societal, social media can more quickly identify how near shore marine resource use adapts. As climate change threatens additional economic and societal disturbances, it will be necessary for resource managers to collect reliable data efficiently to better target monitoring and management efforts.

## INTRODUCTION

In the absence of comprehensive monitoring data, the use of all available data to identify trends in non-commercial fisheries is critical to enable resource managers to make informed decisions, particularly during times where conditions may be in flux (*Ashford et al., 2010*; *Belhabib et al., 2016*; *Bradley et al., 2019*). Social media offers the opportunity to characterize fisher behavior and catch composition across broad spatial and temporal

scales in a timely and cost-effective manner (*Gutowsky et al., 2013*; *Monkman, Kaiser & Hyder, 2018*; *Bradley et al., 2019*). Data mining social media sites have proven to provide reliable and accurate species composition, seasonality of effort, spatial distribution of effort, and basic life history data from over 70 applications in non-commercial fisheries worldwide (*Bradley et al., 2019*). For example, *Belhabib et al. (2016)* used social media to document increases in West African recreational fisheries, most of which was unreported in the official fisheries statistics. In the Mediterranean, scholars have used videos available on social media to identify the types and quantities of fish caught recreationally to aid in creating a profile of the fishery (*Bulleri & Benedetti-Cecchi, 2014*; *Giovos et al., 2018*). In particular, social media sources hold promise for providing real-time data on poorly monitored non-commercial fisheries that could allow fisheries managers to effectively respond during large scale disruptions. However, it is also worth mentioning that previous studies using social media data have paid little attention to the Pacific Island region where non-commercial fisheries are particularly data sparse.

The global COVID-19 pandemic is a good example of a large-scale disruption that has impacted fisheries globally (*NOAA, 2020*; *White et al., 2020*; *Pita et al., 2021*). Impacts to commercial fisheries have been documented in the continental United States (*Smith et al., 2020*), and around the world (*Villasante et al., 2021*). The pandemic's impact on recreational and subsistence fisheries is still poorly understood; however, evidence suggests there was an increase in fishing licenses (*Recreational Boating & Fishing Foundation (RBFF), 2020*) and people fishing across the United States (*Recreational Boating & Fishing Foundation (RBFF), 2020*; *Midway et al., 2021*) and in areas of the Pacific (*Wilson, 2020*), including Hawai'i (*Hawai'i Division of Aquatic Resources (DAR), 2020a*; *Ladao, 2021*). In many localities, non-commercial fishers, which include both recreational and subsistence fishers, outnumber commercial fishers and are often well equipped, highly mobile, and extremely efficient at finding and catching fishes (*Arlinghaus et al., 2019*). For example, an estimated 118 million people across North America, Europe, and Oceania participate in non-commercial fisheries (*Arlinghaus, Tillner & Bork, 2015*). The COVID-19 pandemic resulted in large-scale economic disruptions which included rampant unemployment across large portions of the world, closures, and lockdowns (*Financial Times, 2022*). Economic hardship resulting from unemployment may incentivize increased levels of fishing in order to maintain food security (*Wilson, 2020*; *Ladao, 2021*). Furthermore, increases in available time and a lack of other recreational options could also prompt recreational fishers to spend more time fishing and induce newcomers to enter non-commercial fisheries (*Ladao, 2021*; *Midway et al., 2021*; *Peterson, 2021*; *Recreational Boating & Fishing Foundation (RBFF), 2020*). While these changes in non-commercial fisheries are suspected of being potentially highly impactful to the populations of targeted species, there is little direct evidence to determine how non-commercial fisheries may be responding to the broader impacts of the COVID-19 pandemic.

Despite their large numbers of participants and potential ecological impact, data on non-commercial fisheries are often sparse due to the challenge of accurately characterizing catch and effort in these large and often dispersed fisheries without strict reporting requirements, especially in Hawai'i where local fishers are not required to purchase fishing

permits (*Carter, Crosson & Liese, 2015*). The Hawai'i Division of Aquatic Resources relies on self-reported data from phone and from field-intercept surveys (Study Group on the *Feasibility of a Non-Commercial Fishing Registry, Permit or Licensing System in Hawai'i, 2016*). Self-reported data can have a recall error bias of underreporting or overreporting depending on the recall length (*Chu et al., 1992*), are resource intensive, costly, and are slow to analyze, limiting the ability of managers to respond to rapid changes. Therefore, as a result of this data challenge, alternative methods for obtaining catch data, fishing preferences, and species information are increasingly common, especially when fishery conditions are suspected to be rapidly changing (*e.g., Ashford et al., 2010, Carter, Crosson & Liese, 2015*; *Belhabib et al., 2016*; *Dowling et al., 2016*; *Hartill et al., 2016*; *Banha et al., 2017*; *Giovos et al., 2018*; *Gundelund et al., 2020*).

Hawai'i represents an ideal case to explore the effects of COVID-19 on nearshore fisheries. Anecdotal reports from personnel at the Hawai'i Division of Conservation and Enforcement (DOCARE) and the Hawai'i Division of Aquatic Resources (DAR) suggest that nearshore environments are under increased fishing pressure since the start of the COVID-19 pandemic. The nearshore non-commercial fishery in Hawai'i is critical as both a source of food security and recreational opportunities for a large proportion of state residents (*Kittinger et al., 2015*; *Delaney et al., 2017*; *McCoy et al., 2018*). While the exact size of the non-commercial fishery is unknown, it is estimated that anywhere from 84,000–261,000 residents participate in the fishery any given year (*NOAA, 2012*; National Marine Fisheries Service, Fisheries Statistics Division, 2022, personal communication) which makes it considerably larger than the commercial fisheries operating around the Hawaiian Islands (*McCoy et al., 2018*). Catch and effort data for the nearshore non-commercial fishery in Hawai'i are scarce; however, anecdotal data suggest that fishing activity increased significantly during the COVID-19 pandemic (*Ladao, 2021*). Certainly, the conditions, such as changes in unemployment rates or activities allowed under stay-at-home orders, that might result in increases in non-commercial fishing activity were present throughout the pandemic. The unemployment rate in Hawai'i peaked at 22% (*USBLS, 2022a*) and remained above the national average, ranking at either the highest or second highest state unemployment rate throughout much of 2020–2021 (*Mak, 2021*; *USBLS, 2022a*, *2022b*). Furthermore, lockdown measures intended to limit the transmission of COVID-19 resulted in fishing being one of the few shore-based activities allowed during much of this period (*Hawai'i Division of Aquatic Resources (DAR), 2020b*). However, without adequate data there is a genuine risk that management actions could fail to meet the needs of constituents or miss critical changes in catch or effort that could adversely impact the sustainability of the fishery.

Therefore, our objective was to quantify changes in the nearshore non-commercial fishery around the island of Hawai'i by mining social media data and collecting oral histories from fishers. Hawai'i is the largest of the Hawaiian Islands and is home to approximately 200,000 people (*U.S. Census Bureau, 2021*) with an additional 148,000 ± 21,000 visitors per month on average (± SD; *Hawai'i Tourism Authority, 2022*). A

significant proportion of the people living on or visiting Hawai'i Island consume fishes captured as part of the non-commercial nearshore fishery. The most recent data available reported at the level of individual islands suggest that 10.5–15.1% of the 63,000–65,000 households on Hawai'i Island participate in the non-commercial nearshore fishery, each averaging (± SD) 10.5 ± 2.7 fishing trips per 2-month reporting period (*McCoy et al., 2018*). A large proportion of these households that participate in the nearshore non-commercial fishery likely rely on their catch as an important component to ensuring food security (*Kittinger et al., 2015*; *Delaney et al., 2017*). The majority of fishes captured are for personal consumption within a household or extended family, but there can be significant movement of fish across the island as portions of a catch are given, traded, or sold (*Kittinger et al., 2015*; *Delaney et al., 2017*). An unknown number of visitors also partake in fishing activities that would be considered part of the non-commercial nearshore fishery. The non-commercial nearshore fishery in the Hawaiian Islands includes pelagic species, such as tunas and billfishes; coastal pelagic species, such as carangids; deepwater bottom fishes, primarily lutjanid snappers; and a wide variety of reef fishes (*McCoy et al., 2018*). While angling is probably the most widely used method, castnets, spears, and gill nets, referred to locally as lay nets, are also commonly employed depending upon the species targeted or location fished (*Kittinger et al., 2015*; *Delaney et al., 2017*).

## MATERIALS AND METHODS

### Study area

We focused our survey of social media posts on those geotagged to the island of Hawai'i. However, due to COVID-related restrictions, our one-on-one interactions with fishers were restricted to the shoreline and boat ramp around Hilo Bay, a 31-km$^2$ embayment located on the east side of Hawai'i Island and surrounded by the city of Hilo, the largest urban area on the island with a population over 45,000. Data for the non-commercial fishery of Hilo Bay are not separated from that for the rest of the island during reporting. However, there is no reason to suspect that it differs substantially from the non-commercial fishery as a whole for Hawai'i Island in terms of targeted species, gear types used, or fisher behavior. However, there are likely regional differences around the island in terms of the relative proportion of the targeted species landed and the relative proportion of gear types used.

### Social media data collection and analysis

We restricted our examination to publicly available Instagram, hereafter referred to as social media, posts containing images of captured fishes that were georeferenced to Hawai'i Island and posted between 01 January 2016–05 July 2021. The hashtags of images georeferenced to Hawai'i Island were then used to restrict the sample to images that included "fishing" in the hashtag. Images posted by charter fishing companies were readily identifiable and excluded from our dataset. Posts from individuals that indicated a place of residence other than Hawai'i Island on their profile were also excluded, as were posts

containing images that contained cooked fish or fish otherwise prepared for consumption, *e.g.*, uncooked fish filets. Once images were selected, we enumerated all fishes in the image and identified them to the lowest taxonomic unit possible. The gear type used to capture the fish was either determined from the hashtag, *e.g.*, "#spearfishingbigisland," or based on evidence in the image, such as the presence of wounds indicative of capture in a castnet or being speared, presence of the gear in the image, *etc*. Images posted prior to 23 March 2020, *i.e.*, the day stay at home orders went into effect in Hawai'i, were categorized as pre-pandemic, while those posted after 23 March 2020 were categorized as having been taken during the COVID-19 pandemic and associated lockdowns. The pictured species were categorized as pelagic, coastal pelagic, or reef fishes based on habitat preferences described in *Randall (2007)*.

We used a zero-inflated (Poisson distribution) mixed-model to evaluate whether the mean number of fishes or species of fishes were different before and during the COVID-19 pandemic. Time period (pre-pandemic *vs.* pandemic) and gear type (angling, castnet, spear) were used as fixed effects in both models, while year was treated as a random effect. We evaluated whether there were differences in the species composition of the fishes, grouped as pelagic, coastal pelagic, and reef fishes, in photos posted pre-pandemic or during the pandemic using a chi-square test for independence. All statistical analyses were performed using SAS v. 9.4 (SAS Institute, Cary, NC, USA) at a significance level of $\alpha = 0.05$.

## Collection and analysis of fishers' experiences

We visited 15 locations along the western and southern shoreline of Hilo Bay that allowed public access to the water. Sites were visited a minimum of four times (weekday day, weekday evening, weekend day, and weekend evening) to attempt to represent the full range of fishers using Hilo Bay. We approached groups or individual fishers, identified ourselves, and explained the study and its purpose. We then asked if the fisher(s) would like to participate by providing a brief oral history of their experiences with the COVID-19 pandemic as it related to their interaction with the fisheries resources of Hilo Bay. Verbal consent was obtained, and all data were anonymized. We developed a set of prompts to encourage the fishers to include a brief discussion of how the COVID-19 pandemic had affected the species they targeted, the gear used, their fishing motivation (recreational only, subsistence only, or a combination), and their feelings and attitudes, *i.e.*, about the pandemic, fishing during lockdown, *etc.*, as part of their oral histories. Researchers wrote down responses as the fisher provided their oral histories, then combined their notes after the conclusion of the session. In total, the entire process with each fisher took approximately 15 min.

Responses were coded for analysis as either yes/no, increase/decrease/no change, or assigned to categories, *e.g.*, fishing motivation. We evaluated the interaction of changes in hours worked during the pandemic, reliance upon catch, fishing motivation, and changes in fishing effort or species targeted using chi-square tests for independence. As above, statistical analyses were performed using SAS v. 9.4 at a significance level of $\alpha = 0.05$.
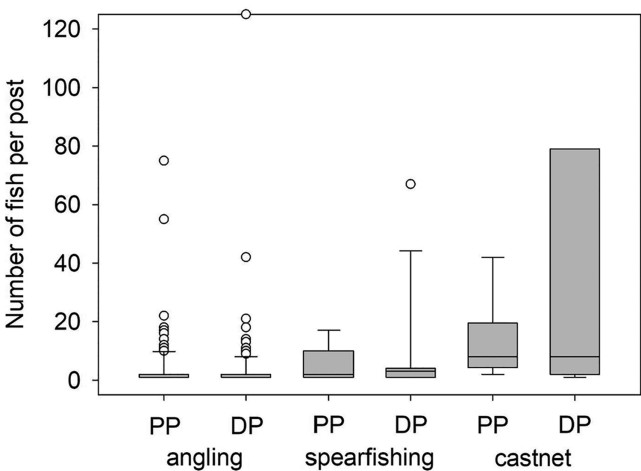

**Figure 1 Number of fish per post by fishing gear type.** Number of fish per photograph separated by gear type in photographs publicly posted to Instagram and geotagged to the island of Hawai'i prior to the COVID-19 pandemic (2016–2019; PP) and during the pandemic (2020–2021; DP). The mean number of fish per post captured by spearfishing and castnet increased during the pandemic relative to the pre-pandemic periods ($F_{1,289} \geq 9.01$; $P \leq 0.01$), while there was no change in the mean number of fish per post captured by angling ($F_{1,289} = 2.49$, $P = 0.12$). The State of Hawai'i officially entered lockdown status in response to COVID-19 on 23 March 2020.

## Ethics statement

This study is exempt according to the Code of Federal Regulations at 45 CFR 46. 104(d) 3 and was approved by the Institutional Review Board (IRB) of the University of Hawai'i.

## RESULTS

### Social media

We collected data from a total of 302 publicly available photographs uploaded to social media by 150 users during 01 January 2016–05 July 2021. The number of photographs were approximately evenly distributed between the pre-pandemic ($n = 147$) and pandemic ($n = 155$) periods. Each user uploaded a mean (± SD) of 2 ± 1 posts and the posting rate during the pandemic period (2.2 posts/week) was almost three times of that during the pre-pandemic period (0.7 posts/week). Approximately 85% of the photographs showed fishes that were captured by angling, with the remainder of posts featuring fishes captured by spearfishing (10%) or cast net (5%). There was no evidence that the frequency of posts from users employing different gear types varied between the pre-pandemic and pandemic periods. However, approximately 90% of the photographs pictured ≤3 fishes, and 75% showed only a single fish.

The number of fishes pictured per post was, on average, approximately 50% greater during the pandemic than the pre-pandemic period (Fig. 1. $F_{2,89} = 8.15$, $P = 0.01$). However, the gear type used by the poster was an important factor in determining whether there were differences in the number of fishes pictured per post between the pandemic and pre-pandemic period ($F_{3,289} = 168.84$, $P = 0.01$). Fishers using spears ($F_{1,289} = 9.01$; $P = 0.01$) or cast nets ($F_{1,289} = 24.20$; $P = 0.01$) posted photographs with a greater number

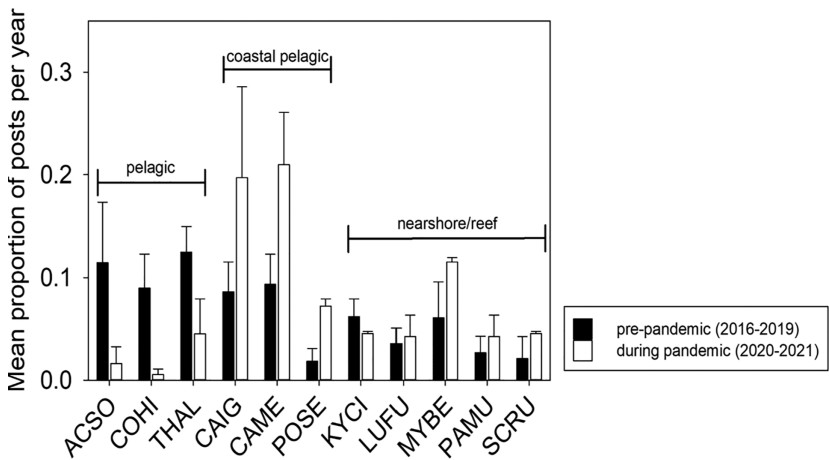

**Figure 2 Mean proportion of Instagram posts per year by top fish species.** Mean (± SE) proportion of Instagram posts per year by the top 11 species of targeted fishes appeared in social media posts geotagged to the island of Hawai'i prior to the COVID-19 pandemic (2016–2019) and during the pandemic (2020–2021). There were three species of pelagic fishes: Wahoo (Ono) *Acanthocybium solandri* (ACSO), Mahi Mahi *Coryphaena hippurus* (COHI), and Yellowfin Tuna (Ahi) *Thunnus albacares* (THAL); three species of coastal pelagic fishes: Giant Trevally (Ulua Aukea) *Caranx ignobilis* (CAIG), Bluefin Trevally ('Omilu) *Caranx melampygus* (CAME), and Pacific Threadfin (Moi) *Polydactylus sexfilis* (POSE); and five species of nearshore/reef fishes: Highfin Chub (Nenue) *Kyphosus cinerascens* (KYCI), Blacktail Snapper (To'au) *Lutjanus fulvus* (LUFU), Bigscale Soldierfish ('U'u) *Myripristis berndti* (MYBE), Manybar Goatfish (Moano) *Parupeneus multifasciatus* (PAMU), and Ember Parrotfish (Uhu) *Scarus rubroviolaceus* (SCRU); comprising the most frequently occurring species. The State of Hawai'i officially entered lockdown status on 23 March 2020.

of fishes pictured during the pandemic period relative to photographs posted during the pre-pandemic period (Fig. 1). In contrast, there was no difference between the pre-pandemic and pandemic periods in the mean number of fish pictured per post by anglers ($F_{1,289} = 2.49$, $P = 0.12$; Fig. 1).

There was no difference between the pre-pandemic and pandemic periods in the mean number of species appearing in a post across the different gear types ($F_{3,289} = 0.91$, $P = 0.44$). However, the species composition of the fishes pictured in the photographs posted to social media did shift to being dominated by coastal pelagic and reef species during the pandemic from pelagic species during the pre-pandemic period ($\chi 2 = 50.33$, df = 10, $P = 0.01$; Fig. 2). Prior to the pandemic, three pelagic species, Yellowfin Tuna (Ahi) *Thunnus albacares*, Wahoo (Ono) *Acanthocybium solandri*, and Mahi Mahi *Coryphaena hippurus*, together appeared in 44.9% of the social media posts. However, these same three pelagic species were only in 7.9% of the photographs posted during the pandemic period (Fig. 2). In contrast, coastal pelagic species, *e.g.*, Giant Trevally (Ulua Aukea) *Caranx ignobilis*, Bluefin Trevally ('Omilu) *Caranx melampygus*, and Pacific Threadfin (Moi) *Polydactylus sexfilis*, appeared in 57.5% of photographs posted during the pandemic period compared to 27.1% before the pandemic (Fig. 2). The overall increase in reef fish occurrence in photographs posted during the pandemic (34.6%) relative to the pre-pandemic period (28.0%) was more modest than the changes observed for pelagic or coastal pelagic species (Fig. 2).

**Table 1 Percent of recreational, subsistence, and recreational/subsistence fishers participating in the noncommercial fishery of Hilo Bay, Hawaii.** Percent of recreational, subsistence, and recreational/subsistence fishers participating in the noncommercial fishery of Hilo Bay, Hawaii reporting an increase (+), decrease (−), or no change (NC) in work hours, fishing activity, reliance on their catch for food security, and a change (C) or no change in species targeted after the start of the COVID-19 pandemic and associated lockdowns in Hawai'i on 23 March 2020.

|  |  | Work hours | | | Fishing activity | | | Reliance on catch | | | Species targeted | |
|---|---|---|---|---|---|---|---|---|---|---|---|---|
|  | **n** | **+** | **−** | **NC** | **+** | **−** | **NC** | **+** | **−** | **NC** | **NC** | **C** |
| Recreation only | 19 | 5.2% | 15.8% | 79.0% | 5.3% | 26.3% | 68.4% | 0.0% | 0.0% | 100.0% | 100.0% | 0.0% |
| Subsistence only | 8 | 25.0% | 25.0% | 50.0% | 25.0% | 12.5% | 62.5% | 50.0% | 0.0% | 50.0% | 62.5% | 37.5% |
| Both recreation and subsistence | 42 | 19.1% | 19.1% | 61.8% | 38.0% | 42.8% | 19.1% | 26.2% | 0.0% | 73.8% | 90.5% | 9.5% |
| Overall | 69 | 18.8% | 15.9% | 65.3% | 27.5% | 34.8% | 37.7% | 21.7% | 0.0% | 78.3% | 89.9% | 10.1% |

### Fisher experiences

A total of 69 fishers participating in the Hilo Bay non-commercial fishery agreed to share their experiences of how their fishing behavior was affected by the COVID-19 pandemic and associated lockdowns that started on 23 March 2020 in Hawai'i. Most of these fishers were angling ($n = 63$) when they were approached, but a relatively small number of individuals were spearfishing ($n = 3$) or using a cast net ($n = 3$). However, approximately 52% of fishers mentioned using two or more fishing methods. Most fishers ($n = 42$) reported that their fishing was for both recreation and subsistence; frequently mentioning that they fished mostly for fun but ate what they caught. It was less common for fishers to indicate fishing exclusively for recreational ($n = 19$) or subsistence ($n = 8$) purposes.

The response of fishers to the COVID-19 pandemic and associated lockdowns, in terms of the amount of time they spent fishing, their reliance upon their catch for food security, and changes in species targeted differed depending upon their stated primary motivation for fishing, *e.g.*, subsistence, recreation, *etc.* (Table 1). However, fishers' reported change in work hours after the start of the COVID-19 pandemic and associated lockdowns was unrelated to their stated motivation ($\chi^2 = 3.18$, df = 4, $P = 0.53$). Despite this, individuals who fished for subsistence exclusively or both for recreation and subsistence were more likely to report an increase in fishing activity ($\chi2 = 17.41$, df = 4, $P = 0.01$) and an increase in reliance upon their catch ($\chi^2 = 9.52$, df = 2, $P = 0.01$) since 23 March 2020 than individuals who fished exclusively for recreation. Further, individuals fishing exclusively for subsistence were more likely to report fishing for different species during the pandemic than individuals fishing recreationally to some degree ($\chi^2 = 8.73$, df = 2, $P = 0.01$).

While there was no relationship between fishing motivation and changes in work hours during the pandemic, fishers who reported a decrease in hours worked during the pandemic were more likely to report an increase in fishing effort ($\chi^2 = 22.86$, df = 4, $P = 0.01$; Table 2). However, there was no relationship between reported changes in hours worked during the pandemic by fishers and changes in their reliance upon their catch for food security ($\chi^2 = 3.02$, df = 2, $P = 0.22$; Table 2).

**Table 2 Percent of fishers in the non-commercial fishery of Hilo Bay, Hawai'i reporting changes in fishing habits by changes in work hours.** Percent of fishers participating in the non-commerical fishery of Hilo Bay, Hawaii reporting an increase (+), decrease (−), or no change (NC) in work hours that reported changes in their fishing activity and reliance on their catch for food security after the start of the COVID-19 pandemic and associated lockdowns in Hawai'i on 23 March 2020.

| Work hours | n | Fishing activity | | | Reliance on catch | | |
|---|---|---|---|---|---|---|---|
| | | + | − | NC | + | − | NC |
| + | 13 | 7.7% | 61.5% | 30.8% | 30.8% | 0.0% | 69.2% |
| − | 11 | 81.8% | 9.1% | 9.1% | 36.4% | 0.0% | 63.6% |
| NC | 45 | 20.0% | 33.3% | 46.7% | 15.6% | 0.0% | 84.4% |
| Overall | 69 | 27.5% | 34.8% | 37.7% | 21.7% | 0.0% | 78.3% |

## DISCUSSION

Since the beginning of the COVID-19 pandemic in Hawai'i during March 2020, the state consistently maintained one of the lowest infection and mortality rates in the U.S. by imposing strict lockdowns and restricting visitors. Concurrently, the state's economy was severely impacted by the decline in tourism revenue resulting in the highest unemployment rate in the U.S. (*Centers for Disease Control and Prevention (CDC), 2021*; *USBLS, 2022a*). The unemployment rate in Hawai'i peaked at 22% (*USBLS, 2022a*) and remained above the national average, ranking at either the highest or second highest state unemployment rate throughout much of 2020–2021 (*Mak, 2021*; *USBLS, 2022a*, *2022b*). Furthermore, lockdown measures resulted in fishing being one of the few shore-based activities allowed during much of this period (*Hawai'i Division of Aquatic Resources (DAR), 2020b*). With many Hawai'i residents facing economic hardships or simply more free time and fewer options available for outdoor recreation, our data indicated an increased reliance on nearshore fisheries resources for both food and recreational opportunities. However, the use of two independent approaches, *i.e.*, catch pictures posted to social media and oral histories, showed that there is considerable nuance and complexity within this broad conclusion that would not have been captured had only a single approach been applied. For example, both approaches indicated that fishing effort was greater during the pandemic than prior to it. Oral history data suggested that increases in effort were being driven primarily by individuals who saw reductions in work hours after 23 March 2020, regardless of whether their primary motivation for participating in the non-commercial fishery was subsistence, recreation, or a combination of the two, which corresponds with oral history data from across several Pacific Islands, including Hawai'i, collected during the pandemic (*Kleiber et al., 2022*). Previous studies suggest that participation in recreational fisheries is negatively correlated to unemployment rate (*Arlinghaus, Tillner & Bork, 2015*), likely due to financial constraints (*Floyd & Lee, 2002*; *Arlinghaus, 2006*). However, these previous studies have focused on factors influencing participation across considerably longer time scales. Participation in the non-commercial fishery in Hawai'i was widespread prior to the pandemic (*Delaney et al., 2017*; *McCoy et al., 2018*) meaning that there were likely few financial barriers for current fishers to increase

their participation as described in previous studies (*Floyd & Lee, 2002*; *Arlinghaus, 2006*). The cost to individuals entering the fishery after the start of the pandemic was unlikely to be prohibitive given the ease of access to the shoreline in Hawai'i, the relatively low cost of gear, and the lack of other recreational activities to spend money on during the pandemic. However, the degree to which financial considerations hindered individuals from entering the fishery after March 2020 is strictly conjecture, as we did not speak to anyone who had entered the fishery during the pandemic. Furthermore, the increase in effort seen in the nearshore fishery in Hawai'i was atypical (see *Midway et al., 2021* for an exception) when compared to trends in other recreational fisheries around the world (*Pita et al., 2021*). Unsurprisingly, increased effort in the nearshore non-commercial fishery in Hawai'i seems to have resulted in increased catch as evidenced by an overall increased rate of posting photographs to social media and an increase in the number of fish per post. The number of fish per post, which was assumed to represent the entire catch for a fishing trip, was being driven by fishers employing spears and cast nets and no corresponding increase was seen in the number of fish per post from anglers. Active gears, like spears and cast nets, tend to have a higher catch per unit effort than passive methods, such as angling (*Dalzell, 1996*), and this efficiency may make them preferred gear types for individuals relying on their catch for food security.

Combined, the oral histories and catch photos posted to social media suggest substantial shifts in the motivation of individuals participating in the nearshore fishery Hawai'i during the initial year of the COVID-19 pandemic. A substantial proportion of individuals who identified as fishing primarily or partially for subsistence indicated an increased reliance upon their catch for food security during the pandemic. However, this increased reliance on catching fishes was reported at a lower rate than that seen in more widespread studies of changes in reliance upon home food procurement in households in Vermont (*Niles et al., 2021*). Furthermore, increased reliance on fishing for food security did not seem to be directly related to work hours, as individuals reporting decreases in work hours increased their reliance upon their catch for food security at about the same rate as those who reported working more hours during the COVID-19 pandemic in Hawai'i. This contrasted with individuals in Vermont that reported a negative job change being more likely to increase participation in fishing activities (*Niles et al., 2021*). These differences between participants in the nearshore fishery in Hawai'i and those in Vermont during the COVID-19 pandemic may have to do with differences in the degree of dependence upon the fishery for food security prior to the pandemic. Hawai'i residents are generally considered to be more reliant on fishery resources for food security, cultural practices, and recreation (*Grafeld et al., 2017*) than those of any other state outside of Alaska (*Loring, Gerlach & Harrison, 2013*; *Harrison & Loring, 2016*) and consume 80% more seafood than the US average (*Grafeld et al., 2017*).

In addition to an increased reliance upon their catch for food security during the pandemic, data gathered both from pictures posted to social media and relayed through oral histories provided by fishers indicated that different species were targeted during the pandemic. Particularly notable is the decrease in the occurrence of pelagic species, such Yellowfin Tuna (Ahi), Mahi Mahi and Wahoo (Ono), a corresponding increase in the

occurrence of coastal pelagic species, such as Giant Trevally (Ulua Aukea) and Bluefin Trevally ('Omilu), and reef fishes. This shift was somewhat inevitable as these pelagic species, as well as istiophorid billfishes, are the primary targets of the charter boat fishery that is heavily focused on serving tourists. However, social media accounts representing charter fishing companies were readily identified and removed from our dataset so the observed decline in the occurrence of pelagic species likely reflected a change in the fishing patterns of Hawaiian residents, but it is unclear what the underlying change was.

For example, the decline in the occurrence of pelagic fishes in catch pictures posted to social media could indicate that fishers with small boats curtailed their participation in the non-commercial fishery. As part of the lockdown orders issued by the state, restrictions were placed on the number of people allowed on small boats, reducing capacity to two people unless they shared the same home address. This capacity restriction, along with the fact that offshore fishing can be expensive due to the costs of fuel, bait, and other tackle, may have contributed to the shift away from pelagic species during the initial year of the COVID-19 pandemic. Alternatively, the decreased occurrence of pelagic fishes in catch pictures could indicate a shift in effort of these small boat fishers or an increase of shore fishers. The coastal pelagic and reef species whose occurrence increased during the pandemic are more readily captured from the shoreline, which is not only less expensive than traveling farther offshore to target pelagic species, but also represented one of the few opportunities Hawai'i residents had to linger on the shoreline when pandemic-related restrictions were in place.

The combination of increased effort and a potential shift of fishing effort to coastal pelagic and reef species is likely to have severe implications to their sustainability, as the reliance of Hawai'i residents on nearshore fisheries already places a heavy strain on nearshore ecosystems under normal conditions. For example, the nearshore non-commercial fishery provides an estimated seven million meals per year in Hawai'i and most of the targeted species, for which data exist, have shown signs of overfishing (*Nadon et al., 2015*; *Grafeld et al., 2017*; *McCoy et al., 2018*). Similar changes in fishing patterns associated with COVID-19 that vary across socio-economic groups within a fishery have been documented elsewhere (*Smith et al., 2020*; *Wilson, 2020*).

Our results suggest that mixed recreational and subsistence fisheries require management that focuses on ensuring resiliency to guarantee that the resource can sustain periods of elevated exploitation that might follow socio-economic disruptions particularly since disruptions on the scale of the global COVID-19 pandemic are predicted to increase in frequency in response to changing climatic conditions (*Costello et al., 2009*; *IPCC, 2019*; *USGS, 2022*). Whereas effects from the pandemic unfolded rapidly, climate change itself will occur over a much longer time scale (*Fuentes et al., 2020*; *Leal-Filho, Nagy & Ayal, 2020*). Many of its impacts, however, are expected to be rapid, abrupt, and encompass non-linear shifts (*IPCC, 2018*, *2019*; *Trisos, Merow & Pigot, 2020*) with some tipping points predicted to occur with only a 1.5–2.0 °C of average global warming (*IPCC, 2018*). These impacts, along with the increase in frequency and intensity of hydrologic and climatic natural disasters, will have compounding and cascading effects (*AghaKouchak et al., 2018*; *Phillips et al., 2020*) that, like during the pandemic, will exacerbate existing economic

stressors, especially for vulnerable populations including the unhoused, low-income families, and native Hawaiians (*Hawai'i State Office of Planning, 2021*). For instance, climate change will mimic the pandemic in causing multiple impacts that will disrupt global supply chains (*Kovács & Sigala, 2021*) and thus increase food insecurity, especially in Hawai'i as more than 85% of food is imported (*Hawai'i State Office of Planning, 2021*). These changes will have significant impacts on income, livelihoods, and food security of marine resource-dependent communities throughout the Pacific (*Lam et al., 2016*). Moreover, ocean warming and acidification, combined with other anthropogenic stressors, are predicted to strongly affect coral reef fish communities (*IPCC, 2019*), resulting in shifts in species distributions, declining fish populations, and lower catch rates. In addition, these changes will result in fishers continuing to alter their fishing effort, fishing location, and species targeted (*Hanich et al., 2018*). Further, our results suggest that fisher behavior during a disruption may be different than expectations based upon their behavior before a disruption, such as the species targeted in a multi-species fishery. The *IPCC (2019)* estimates a 20–25% decline in global fisheries catch within the foreseeable future, increasing tensions between fisheries sustainability, natural resource managers, and communities.

The costs of traditional data collection methods will continue to hinder the ability for resource managers to collect data in an efficient manner to adapt effective management strategies. This limitation presents an opportunity for researchers and managers to find new sources of data, like social media, that can provide reliable information in a timely manner. It should be noted that like all fisheries data, data collected from social media possesses biases; however, social media and online angler apps have been demonstrated to produce accurate, repeatable, and cost-effective fisheries data (*Papenfuss et al., 2015*; *Belhabib et al., 2016*; *Banha et al., 2017*; *Sbragaglia et al., 2019*). Based on studies using social media data collected from other platforms, catch pictures on the social media platform we used are likely to have been posted by fishers who are younger and have a greater degree of online access than the fisher population as a whole (*Papenfuss et al., 2015*). Further, relying on catch pictures will likely underestimate effort as there is a lack information on fishing events that resulted in no catch (*McCluskey & Lewison, 2008*; *McCormick, Quist & Schill, 2013*). Despite these potential sources of bias, our results indicate that social media and traditional data collection methods showed similar trends in the non-commercial nearshore fishery in Hawai'i. While they are unlikely to supplant traditional data collection methods, such as creel surveys, developing sampling protocols that allow data mining of publicly available data of fishers' behavior may offer a complementary approach to monitoring fisheries that could allow for more rapid detection and response to changes driven by large-scale disruption given the time and resource intensive nature of more traditional monitoring methods.

## CONCLUSIONS

This article explores how a global shock affects nearshore fisheries in Hawai'i. There are three main takeaways from this study. During the pandemic: (1) resource users posted nearly three times as often with nearly double the number of fishes pictured per post;

 

(2) individuals who fished for subsistence were more likely to increase the amount of time spent fishing and relied more on their catch for food security; and (3) individuals fishing exclusively for subsistence were more likely to fish for different species. The oral histories we collected from fishers validate our social media findings, suggesting that social media data can be used to rapidly collect data and predict changes in nearshore fisheries because of large-scale disturbances. As climate change threatens additional disturbances, it will be necessary for resource managers to collect reliable data quickly to prevent unsustainable fishing pressures and to better target management plans.

## ACKNOWLEDGEMENTS

P. Maurin provided comments and suggestions that improved this manuscript.

The Hawai'i Cooperative Fishery Research Unit is jointly sponsored by the U.S. Geological Survey, the University of Hawai'i System, the Hawai'i Department of Land and Natural Resources, and the U.S. Fish and Wildlife Service. Use of trade, firm, or product names is for descriptive purposes only and does not imply endorsement by the U.S. Government.

### Funding

This work was supported by an NSF-Research Experience for Undergraduates site grant (#1757875, Rebecca Ostertag and Noelani Puniwai) through the Pacific Internship Programs for Exploring Science based at the University of Hawai'i (UH) at Hilo.

The Hawai'i Cooperative Fishery Research Unit is jointly sponsored by the U.S. Geological Survey, the University of Hawai'i System, the Hawai'i Department of Land and Natural Resources, and the U.S. Fish and Wildlife Service. The funders had no role in study design, data collection and analysis, decision to publish, or preparation of the manuscript.

### Grant Disclosures

The following grant information was disclosed by the authors:
University of Hawai'i (UH): #1757875.
U.S. Geological Survey.
Hawai'i Department of Land and Natural Resources.
U.S. Fish and Wildlife Service.

### Competing Interests

The authors declare that they have no competing interests.

### Author Contributions

- Timothy Grabowski conceived and designed the experiments, analyzed the data, prepared figures and/or tables, authored or reviewed drafts of the article, and approved the final draft.
- Michelle E. Benedum conceived and designed the experiments, authored or reviewed drafts of the article, and approved the final draft.

- Andrew Curley performed the experiments, analyzed the data, prepared figures and/or tables, authored or reviewed drafts of the article, and approved the final draft.
- Cole Dill-De Sa performed the experiments, analyzed the data, prepared figures and/or tables, authored or reviewed drafts of the article, and approved the final draft.
- Michelle Shuey conceived and designed the experiments, authored or reviewed drafts of the article, and approved the final draft.

### Human Ethics

The following information was supplied relating to ethical approvals (*i.e.*, approving body and any reference numbers):

This study was approved by the Institutional Review Board (IRB) at University of Hawai'i at Hilo.

### Data Availability

The raw data is available in the Supplemental Files.

### Supplemental Information

Supplemental information for this article can be found online at http://dx.doi.org/10.7717/peerj.14994#supplemental-information.

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
