# Peer review of "Pandemic-driven changes in the nearshore non-commercial fishery in Hawai’i: catch photos posted to social media capture changes in fisher behavior"

_PeerJ, doi:10.7717/peerj.14994_

## Round 0.1 · original submission · Major Revisions

The two reviewers have provided you with thorough and thoughtful reviews. Please pay close attention to the details during your revisions to ensure your re-submission is acceptable.

Reviewer 1 ·

Basic reporting

Throughout the paper, I suggest being careful using “Hawaiian” (e.g., line 96: The Hawaiian nearshore non-commercial fishery – change to – The nearshore non-commercial fishery in Hawaii). This will be interpreted very differently among readers and may not be the appropriate context for using “Hawaiian”.

• Line 58: suggest saying Pacific Island region
• Line 24: suggest removing first sentence as it is redundant with the following sentence
• Line 132: remove “away”
• Line 151: Choosing this hashtag dramatically affects the pool of “samples” and creates a heavy bias
• Line 242: maintain consistency with hyphen/no hyphen or space/no space in “noncommercial”
• The first paragraph of the materials and methods section contains a lot of introduction type material – suggest moving that content into the introduction

Experimental design

There was a NOAA tech memo published this year titled “Pacific islands region fisheries and COVID-19: impacts and adaptations”. The oral history work in this memo covers many of the topics in this proposed paper. I’d encourage the authors to read through this memo, as some of the data from the memo is from Hawaii Island.

Another relevant and important paper that should be considered and cited in this paper: “FEASIBILITY OF A NON-COMMERCIAL MARINE FISHING REGISTRY, PERMIT, OR LICENSE SYSTEM IN HAWAI‘I”.

Validity of the findings

Although unfortunate, dramatic events like a global pandemic provide an excellent opportunity for comparison studies. I agree with the overall finding that there was an increased effort by non-commercial fishers. The oral history information is also extremely valuable. However, I have many reservations with how the social media information was sourced. There are too many unknown factors behind the human behaviors related to posting on Instagram. I would encourage the authors to be careful with their discussion and conclusions related to the social media info – I acknowledge that the oral histories were used to ground truth the social media data, but I feel too much weight is placed on the social media findings. By nature, social media information can be quite biased and skewed. Access to smart phones and wifi is not evenly distributed demographically. Also, only using one hashtag (#fishing), presents an interesting issue as well. I think these caveats or disclaimers should be acknowledged at some point in the paper.

Additional comments

• Line 145: I would disagree with the statement that Hilo does not differ from other places around Hawaii Island. The terrain, bathymetry, demographics, climate, etc are different from Kona.
• Line 148: Is there anyway to prove that’s where the fish came from? Posters sometimes arbitrarily choose locations or choose a different location/island to purposely mislead viewers on where a fish was landed.
• I’m a little concerned with how the pelagic species play a role in this study. Many of the references and information is regarding nearshore fisheries. The small-boat fishery is very different in many aspects compared to nearshore gear types (shore casting, throw net, spear, etc.).

Reviewer 2 ·

Basic reporting

I feel this paper is somewhat lacking in background and context. The authors have sufficiently referenced appropriate literature related to the COVID-19 pandemic and comparing their findings to other studies, but I found myself wondering why this matters. I believe this paper would be much improved by a brief overview of some of the limitations with current data collection for recreational and subsistence fisheries, and what the effect of these limitations is likely to be. The authors describe in one sentence that existing data are sparse, but I think this needs to be expanded upon. Ultimately, better data collection is needed to understand, at a minimum, 1) how many fish are being removed from a population to better assess population status/health/size, and 2) who is fishing, how, where, and for what. The data helps fisheries managers to better manage a fishery to ensure its sustainability and to maximize resource benefits for users/Hawai'i/etc. I'm not suggesting the authors need to describe the goals of fisheries management, but they need to include in the introduction how new data collection methods can improve fisheries assessment and management.
The authors (correctly) describe in the discussion that resource use can change rapidly during events like the pandemic and with climate change. I agree that the data collection methods described here can help managers understand these changes in resource use, but the authors should also note that these are very different types of events. The COVID-19 pandemic was a sudden, shock event where changes occurred very rapidly. With climate change, on the other hand, most (not all) changes occur more slowly. While it's true that new types of data collection methods can help to see trends from climate change, the authors haven't made a convincing case that these rapid methods are necessary to help understand changes related to climate.

Experimental design

The research question is well stated, and the methods are sound.
One question is about the sampling design of the oral histories: is there a sampling bias here from only visiting shoreside locations? Does this bias your sample toward shore-based anglers, and away from boat-based anglers or spearfishers? Please describe whether this does introduce a bias and how it is accounted for, or if not, why not.

The manuscript does not sufficiently describe the knowledge gap being filled. It's clear that there was limited data about changes in fisher behavior during the pandemic, but the authors haven't sufficiently made the case about why it is important to collect data on these changes, and how they impact the fishery from both a biological and socio-economic standpoint.

Validity of the findings

The findings and data seem robust and sound.

Additional comments

Text edits:
Line 268: "both fishing effort" AND...??
Llne 278: delete "has" - at some point the pandemic will be in the past, and so the paper should refer to the pandemic in the past tense.
Lines 277-283: These few sentences just generally describe what happened in Hawai'i during the pandemic, but need to be better connected to the research question. Also, is it fair to modify the first sentence to read "... by imposing strict lockdowns and discouraging (restricting?) visitors"
Line 309: Midway et al found an increase in effort as well, though. Please modify this statement to better reflect the literature

---

## Round 0.2 · accepted · Accept

Your manuscript is substantially improved. Therefore, I have made the decision to accept it. However, the two reviewers still have a few minor revision requests. Please ensure these issues are addressed before submitting your final manuscript.

Reviewer 1 ·

Basic reporting

Line 367-368 (suggested revision): ...individuals participating in Hawaiʻi's nearshore fishery during the initial year...

Line 369 (suggested revision): remove "as"

Line 323-324: "by imposing strict lockdowns and restricting visitors" was added to the end of the sentence. I would recommend providing a citation for this statement or removing.

Experimental design

No further comments, I appreciate the follow-up in the rebuttal.

Validity of the findings

No further comments, thank you for the explanations in the rebuttal.

Additional comments

N/A

Reviewer 2 ·

Basic reporting

The article is generally well written. Some suggestions to improve writing are as follows:
Abstract – Line 28 (and throughout) – “during the pandemic” we are still technically in the pandemic. The timeline needs to be defined better here

Lines 99-100 – “Alternative methods… are increasingly common” – I think ‘common’ is not the right term here. If alternative methods are common, there is no need for more. Do the authors mean these methods are being used more frequently, or that they are increasingly in demand?

Lines 108-110 describe the anecdotal evidence that non-commercial fishing is increasing, Lines 118-119 say essentially the same thing. Revise to make this paragraph less redundant.

Lines 115-116 – ‘considerably larger than the commercial fisheries…’ – specify that this is by number of participants, rather than other measures of the fishery size (effort, catch, etc.)

Lines 174-176 – ‘However’ is used twice



Lines 430-431 – The authors casually mention here that the change in species is a result of people not fishing from charter boats. While charter boats aren’t part of this study, the suggestion that offshore pelagic species could have benefited from harvest pressure release in this time period is at least worthy of mention as a follow up research question

Line 454 – “…associated with COVID-19” – change to “associated with the COVID-19 pandemic” – this is about the pandemic and the resulting public health measures, not the disease itself

Experimental design

No comment

Validity of the findings

The authors should more carefully review the literature about the impacts of the pandemic on recreational fishing activity around the globe. Other authors have also found an increase in activity, and this should be noted.

Specifically: Lines 386-388 – I’m not sure I would characterize the findings here as being atypical, and those of Pita et al as being more typical. There are other studies from this time period that have similarly found an increase in recreational fishing activity, or a mix of increasing and decreasing activity (see: Stokes et al. 2020, Trudeau et al, 2022). Please more thoroughly review the literature on this and describe this with more nuance